# Impact of Personality Traits on Small Charitable Donations: The Role of Altruism and Attitude towards an Advertisement

**Stefanos Balaskas** [ID]**, Aliki Panagiotarou and Maria Rigou** *[ID]

Department of Management Science and Technology, University of Patras, 26334 Patras, Greece; up1031486@upnet.gr (S.B.); apanagiotarou@upatras.gr (A.P.)
* Correspondence: rigou@upatras.gr

**Abstract:** Given the serious humanitarian crises encountered by the modern world, it is more crucial than ever to build a society based on solidarity, compassion, empathy, and a sense of teamwork and cooperation. This research provides insight into how the factors of personality traits, altruistic behaviors, and humanitarian actions can positively influence citizens' behavioral intentions, allowing for a deeper understanding of the motives behind charitable giving. To this end, a study was conducted with 332 Greek respondents, which used a special purpose questionnaire, including the HEXACO-24 questions and 13 additional questions, that addressed attitude towards advertisement, charitable giving, as well as the behavioral intention to donate. The findings add to current research by underlining the relevance of individuals' altruistic character, since our analysis revealed a statistically significant relationship between charitable giving and behavioral intention to donate. All HEXACO personality traits, excluding emotionality, demonstrated a statistically significant positive relationship with the attitude towards advertising, while none of the personality traits exhibited a statistically significant positive relationship with the concept of charitable giving, which calls for further investigation. Our findings also revealed that attitude towards an advertisement had no drastic effect on charitable giving and behavioral intention.

**Keywords:** donation; charity; behavioral intention; HEXACO-24; personality traits; altruism; SEM





## 1. Introduction

Philanthropy, or the provision of services to third parties, differs from mutual aid, which is the dominant social and economic support system for most of the population in industrialized countries [1,2], as the primary recipient of the volunteering is not the group member, but an extended third group, although it should be noted that most people consider philanthropy to include elements of personal achievements and interests [1,3]. Thus, we can say that philanthropy is an attitude; a behavior in which the loving disposition towards others is manifested [3,4]. A philanthropist is someone who experiences internally, and expresses externally, the feeling of love, while a charitable work can be described as anything related to and governed by the specific emotional state. Charities are non-profit organizations (NPOs) that focus on charitable purposes, helping fellow human beings in need, thus serving the public interest and the common good [5,6]. They express bonds of comradely and class solidarity of their members, who either participate financially, or contribute through knowledge and volunteer actions to the realization of the mutual goals, and seek as many members as possible to assist in their cause. Thus, excluding specific professions that require volunteering for practicing/learning reasons; in the general case volunteering is identified with charity to give a new virtue, the love for the fellow man. Funding sources are subscriptions, donations, and sponsorships, mainly independent and financially autonomous, relying exclusively on voluntary contributions. Despite the similarities between the concepts of NPOs and volunteering, the latter continues to inspire more, especially in today's era where individualism dominates [3,7–9].

There are a variety of definitions and types of charity [10,11], with each region's history, politics, religion, and culture having an impact on it. The assumption that one's volunteer acts will be appreciated by the community underpins philanthropic and giving behavior. The desire for charitable giving is a manifestation of social interactions through which individuals seek to improve not only their own, but also the livelihoods of those in need, and address development issues [10,11]. A rising issue in today's society that is evolving in response to the pressures of globalization concerns the funding of NPOs. NPOs receive financing from a variety of sources, including general public donations, government and international organization support, and charitable contributions. Funding NPOs that reflect the perspectives of certain groups can help governments collect all important citizen opinions and experiences. For certain groups, it is a productive and representative approach to voice their views on social concerns, making it simpler for governments worldwide to gather all relevant perspectives and experiences of individuals [8,12]. Governments require this information to establish laws and policies that are successful and do not harm the interests of certain groups of people. The economic crisis had a substantial impact on the activity of non-governmental organizations, decreasing the availability of funds and services at a time when their need was greater than ever. Considering the severe political disenchantment, NPOs are increasingly being called upon to fill the hole left by a diminishing welfare state, and to promote human rights and democratic ideals [9,13]. This crisis has offered fresh opportunities for the non-profit sector.

In this article, we present a study that was conducted to investigate the set of elements that impact people's behavior and susceptibility towards prosocial activities, with an emphasis on charitable donations. We investigate what personality traits lad each person to donate, as well as other socio-political aspects such as the individuals' altruistic tendencies. The study was conducted with 332 Greek respondents, which used a special purpose questionnaire with 37 items comprising the HEXACO-24 questions, and 13 questions that addressed attitude towards advertisement, charitable giving as well as the behavioral intention to donate. The subject of determining whether people are motivated to donate is thoroughly researched in psychology, economics, and related sciences. It is essential to comprehend the factors that motivate people to contribute, both psychologically from the standpoint of personality, and those related to external societal influences (such as attitude towards advertisement and perception of charity marketing). These factors are required for the preservation and upkeep of charity organizations and the causes they serve, but also encourage philanthropic behavior. The article is structured as follows: The next two subsections present an overview of the relevant research on the influence of personality traits on people's decisions to make future donations, as well as the connections they exhibit with the altruistic aspects of human nature and prosocial attitudes. We focus our interest on how consumers perceive charity advertisements by studying how attitudes towards the advertisement impact charitable donations. Section 2 contains a detailed description of the model we developed and tested for our research purposes, followed by the data analysis in Section 3. Section 4 examines and interprets the main findings and indicates relevant limitations. Finally, Section 5 concludes the article and makes recommendations for further research.

### 1.1. Personality Traits and Donations

By definition, the "personality" term refers to "the dynamic organization within the individual of those psychophysical systems that determine his unique adjustments to his environment." [14,15]. In recent years, it has come to our attention that, in addition to the external factors (which can influence an individual, thus composing their behavior and adapting it to social and personal circumstances), individual internal factors that compose the personality are also observed in each person independently, namely personality traits [16,17]. These are indicators that describe "people in relation to behavioral patterns, thoughts, and emotions while are relatively stable over time, differ across individuals and are relatively consistent over situations" [18,19]. The term "personality" is

utilized frequently in common speech, and is acknowledged by both the general public and academic circles to refer to a combination of traits that make each individual unique. Personality questionnaires based on lexical theory are one of the most popular and widely used methods of determining and evaluating human nature, as well as for predicting social behavior over time, with particular attention paid to the effect of personality characteristics on philanthropic behaviors [20–22].

The Five-Factor Model (FFM), also known as Big Five Model, one of the most widely used personality models, piqued the interest of two personality researchers, Paul Costa and Robert McCrae [23,24], who supported the validity of the model in question through empirical studies, providing momentum for conducting personality research studies within the specific context in different cultural contexts and cultures, and in a diverse range of different populations [25]. FFM categorizes an individual's personality using five primary traits, namely extraversion, agreeableness, conscientiousness, neuroticism, and openness to experience. Although the Five-Factor Model is one of the most influential models in the history of personality research and theory, it is not the only option that has been proposed.

The six-factor HEXACO model, a variation of the Five-Factor model (Big Five model), has received a lot of attention lately, since it served as the theoretical foundation for the development of numerous personality evaluation instruments [26–28]. A factor's high and low levels can be determined by its characteristics. The most frequent technique to test one's personality characteristics using HEXACO is to utilize an automated report inventory or an observer's report. Each of the six characteristics is the outcome of a set of questions designed to assess the level of each factor. Each of the six HEXACO components is divided into four "aspects", one for each personality trait, and is measured by the HEXACO-PI-R [28]. This model incorporates the Big Five model's five elements, as well as the honesty-humility factor [29], and it also differs in the neuroticism element that it associates it with the emotionality trait.

Through the inclusion of the honesty-humility factor, the HEXACO model was able to explain the uniqueness of some antisocial criteria, such as psychopathy, Machiavellianism, narcissism, and selfishness, but also prosocial tendencies, such as collaboration [29–31]. In these instances, people with low scores of honesty-humility are more prone to act in a manipulative, unjust, self-enhancing, or exploitative way. On the other hand, individuals who demonstrate high scores of honesty-humility are less likely to actively seek out exploitative circumstances and behave in such a way that might lead to advances in reputation inside a cooperation. Low scores on this trait are also linked to negative personality traits, such as selfishness and manipulative tendencies, which are studied in the psychological theory of the dark triad [31]. Conscientiousness has been related to the accomplishment of one's professional, social, or other commitments, and conscientious people are known for their methodicality and thoughtfulness. People with very high levels of conscientiousness and agreeableness dedicate more time to volunteer activities, and are more likely to make monetary donations [30,32,33]. In terms of altruistic attitude, individuals with high levels of conscientiousness may feel obligated to assist others if they believe it is required, or strive towards exemplary citizenship [34], while in some cases altruistic attitudes can be perceived as an obligation by those whose behavior conforms to prosocial principles [32]. Extraversion, according to this theory, operates within the context of social interaction (such as socializing, mentoring or entertainment). High scores of extraversion are likely to provide social advantages, depending on social or environmental factors (i.e., access to friends, associates, or even partners) and have been directly related to proclivity for volunteering and pro-social activities. Individuals with empathy, propensity toward helpful acts, and high degrees of empathy, are characterized by the personality trait of emotionality [26,27,31]. Despite that, the relationship between this trait and prosocial activities has received little attention, especially regarding charity advertisements that include messages and appeal to empathy [35].

It is evident that a person's personality traits may affect their altruistic attitudes and, consequently, how one behaves across all social contexts. Therefore, how people

perceive and interpret their social environments and experiences can be varied depending on their personality, which is an interesting subject of research. The association between prosocial tendencies and personality traits that result in altruistic or charitable activities has received a lot of attention in academia [17–19,30,36]. Empathy, or the degree to which one's behaviors, sense of responsibility, and motives are directed towards the common good, as well as the degree to which one actively promotes this need to assist individuals within their social circle, are the guiding axes of prosocial behavior [33,36]. In [37], the authors conducted a cross-cultural study to assess the influence and relationship between Big Five, HEXACO traits and gratitude towards God on several aspects of well-being. In the first study, scales for cognitive, psychological, and subjective happiness were examined and correlated with dispositional, religious gratitude and Big Five personality traits of 188 Muslim participants. The results showcased agreeableness to be the main determinant for gratitude, while it had no statistical influence on any of the aspects of well-being, in comparison to dispositional gratitude, which proved to be a predictive factor. In the second study that involved 212 Christian participants, HEXACO's honesty-humility and extraversion traits were significantly correlated with gratitude. Once again, dispositional gratitude plays an important role in predicting subjective well-being and life satisfaction, while gratitude towards God and extrinsic-personal religiosity had a direct and positive relationship. The authors note the significance of the results regarding reciprocal altruism, which is correlated with high scores of agreeableness, while gratitude is influenced by honesty-humility.

Lim et al. in [35], approached the issue of the effectiveness of advertising appeals of nonprofit organizations through a 2 × 2 experimental design. The focus was placed on the impact of HEXACO's personality traits on attitudes towards advertising and intention to donate after being exposed to advertisements on social media, to account for the effectiveness of social media metrics (i.e., likes, comments, etc.). Regression analysis revealed that honesty-humility, emotionality and agreeableness had a statistically significant impact on positive attitudes, while extraversion and conscientiousness were more likely to result in actual donations. Interestingly, none of the personality traits had a direct or indirect effect on advertising appeal types, either through attitudes or donation intentions.

Yarkoni et al. in [38], highlighted the significance of personality traits concerning attitudes towards recipients and donation behavior. In their research design, Analog to Multiple Broadband Inventories (AMBI) was implemented to explore personality, which includes several of the major personality scales. A total of 284 participants were exposed to 16 dynamically generated biographies describing individuals seeking humanitarian aid, while each biography, due to its uniqueness, with the inclusion of a predictive algorithm, was later used to acquire normative attitudes. Agreeableness had a significant impact on all aspects of social evaluation, both for recipient and normative attitudes. In addition, extraversion, openness to experience and conscientiousness had a positive influence on perceived responsibility and likeability, as people exhibit sympathetic tendencies and are more likely to identify and recognize a person's need, or even to distinguish individuals who necessitate and require genuine immediate assistance, which generally results in monetary donations.

As mentioned above, attempts have been made to study the role and cruciality of personality traits and their influence on the outcomes of charity marketing. Personality traits allow researchers to explore human nature and discover behavioral patterns in consumers that will allow for designing effecting strategies and marketing campaigns for non-profit and charity organizations. To this end, the current study utilizes the HEXACO-PI to examine the connection between personality and prosocial/altruistic behavior in response to charity advertising.

*1.2. Charitable Attitudes and Advertisement*

Advertising is one of the most fundamental components of the promotional mix, and is often one of the components of a comprehensive marketing and communications program [39,40]. Advertising methods and techniques are implemented to efficiently spread messages while they are employed not just by corporations or organizations, but also by non-profit organizations, museums, charities, government agencies, or any other type of agency that distributes direct messages about a shared aim [41–43]. Businesses, in collaboration with other organizations, non-profit or non-governmental, can take local initiatives for a cause, invest resources in support of community issues, make monetary or charitable donations, encourage and support volunteerism and human rights, and acquire financial support through grants and charitable activities [41,44,45].

To reconcile conflicting interests and demanding audiences, most corporations are now compelled to establish strategies, support a social cause, and encourage philanthropy. A business, typically a for-profit organization, distributes a proportion or all of its profits to a philanthropic or humanitarian purpose. It is a policy that is often applied to a certain product and for a limited time period. Implementing such a policy guarantees mutual advantage, both for the firm increasing sales of a certain product and for the financial support of the beneficiary NPO [3,7,9]. At the same time, consumers are given the opportunity to support a cause without spending any extra money by purchasing the products. With these initiatives, corporations attempt to create a respectable social profile that people would recognize via their efforts, monetary donations, and charity. To an extent, several corporations have already begun to feel a sense of obligation to pursue charitable giving as a beneficial part of their social image. By sponsoring a "good cause", it enhances its social profile and, as a byproduct, indirectly raises its reputation abroad. The non-profit organization may market its cause and attract consumers who will either fund or be intrigued by learning about the cause [46–49]. However, there is the risk of mistaking social responsibility with marketing activities, which can lead to unfavorable impacts on the firm's efforts to advertise their aim. It has been observed that, in several cases, marketing "cancels" the goal of social work and is viewed as advertising, as the consumer public interprets advertising messages as a capitalistic drive for profit [46–49].

To communicate their message, NPOs rely heavily on the psychological aspects of advertising. They evoke emotions such as sadness, remorse, or fear to manipulate the audience [44,50,51]. Charitable and social advertisements use emotional appeals of shame to convince and urge message receivers to acquire products and donate. Emotional appeals emphasizing guilt, in general, are widely implemented to raise audience attention and thus help messages stand out [51,52]. Consumers that feel guilty are those who understand they have disobeyed social rules and betrayed their "beliefs". Individuals who feel guilty also identify their failure to acknowledge and accept responsibility. Other negative emotions, such as shame, underline other people's values and "beliefs", which define what actions should be taken to act upon a social problem [48,49,53–55]. Shame depends on unfavorable assessments of "third parties", whereas guilt is an internal function based on one's own beliefs. Promoting a message with emotional appeals of guilt, the individual may manage the situation while feeling unpleasant, whereas in fear, the person has little or no power. Guilt appeals elicit sentiments of rage and aggravation, causing the recipient to perceive the message as repulsive, whereas low-guilt appeals hardly maintain the receiver's attention [43,48,54,56]. Arguably, among the most fundamental motives for purchasing a product or service is the prioritization of human needs and a new higher level of satisfaction, which undoubtedly contributes to the development of new inter-personal methods for appealing to the consumer public. Thus, the importance of consumer behavior becomes an integral task, particularly psychological aspects since they are individual, personal impulses that stem from the qualities of each individual's personality traits independently. By determining the role that customer personality signifies, marketers could effectively cultivate a plan of action that is more likely to satisfy consumers' motivations and requirements. The realization of how personality traits can have a significant impact

on future purchase intentions prompted the need to classify those reasons that influence consumer behavioral patterns and attitudes towards advertisements, to the extent that their comprehension and study can contribute to the survival and growth of businesses.

In [57], the authors investigate what drives individuals to donate to charitable causes. To study the factors behind altruistic behaviors and behavioral intentions for charitable giving, in their research model, particular emphasis is placed on religious commitment, attitudes towards advertisement and charitable organizations. The results from the structural equation modelling on 214 participants indicate that there are positive direct effects between attitudes towards charitable organizations and helping others scale with behavioral intentions to donate. While there is no significant impact between attitudes towards helping others and final behavioral intention, interestingly, religiosity proved to have an imperative influence with statistically significant relationships between behavioral intention, and attitudes towards advertising and charitable organizations.

In [58], the authors examined the key factors that influence people's happiness in the context of life satisfaction. In their research models, the authors approach this issue by examining how different dimensions of religiosity affect individuals (intrinsic, extrinsic social/personal) and the mediating role of altruism through charitable giving and volunteering. The results from 3008 Turkish participants showed that both charitable giving and volunteering were not affected from intrinsic religiosity, whilst social and personal religious orientations had a significant impact on charitable giving. This observation enhances the theoretical foundation surrounding people's proclivity for altruism when they incorporate their inclusion and projection in a social group, but also on a personal level with the goal of inner well-being and tranquility. It is worth noting that all aspects of religiosity revealed a direct and positive association with life satisfaction.

We have already discussed how emotional appeal in advertising campaigns influences not just the consumer audience and how the presented message is perceived, but also what impact different emotions have on the final behavioral or purchase decision. In [59], researchers attempted to determine how positive and negative appeals on charity advertisements influence consumers' behavioral intentions to donate though a series of four studies. The first part of their studies was focused on identifying whether positive or negative appeals affect attitudes and donation behaviors in different conceptual cases, while the second part focuses on people's expectations and awareness regarding the displayed charity appeal and its effectiveness on actual donation. Based on the results from the studies, the authors highlight the drastic effect of positive appeals on people's perception of the featured charity organization, as well as how self-perceived belief was mediated by positive appeals and improved the attitude to donate. In contrast, negative appeals had a significant impact on actual behavior to donate, while no statistically significant differences were found between the unfavorable attitude towards the organization and willingness to donate.

## 2. Materials and Methods

### 2.1. Research Model

The current study endeavor, driven by the aforementioned theoretical models and contemporary literature, proposes a comprehensive research model that depicts a multifactorial and complex sequence of interconnections integrating notions and concepts such as altruism and personality, but also the way they inspire individuals to charitable deeds and donations. Based on HEXACO's six-factor model, our approach encompasses elements such as personality traits, the principle of altruism by implementing a charitable giving construct, and attitudes toward charity advertisements. Finally, the dependent variable will consist of the behavioral intention to make future monetary donations. We approach the issue of behavioral intention to donate by studying and including various aspects of the cognitive and psychological human conditions. Personality traits, prosocial behaviors, and philanthropy share similarities in terms of context, assessment, and domain, thus posing a challenge in the survey implementation, which required the careful examination

of phrasing in our constructs. The human mind and behavior represent a multifaceted challenge, considering that they are impacted by a variety of external and internal variables such as biological predispositions, culture, age, and so forth [17,18,30,42,60]. In our endeavors, we established a framework that addresses all the beforementioned concepts by defining the components that are essential, the way they are interconnected, and the sequence of interactions that influence the final outcome. We have adapted our constructs in the context of charitable giving and donor behavior, anticipating that the models' validity will be unaffected despite the adapted context; an assumption that was confirmed by our statistical analysis in Section 3. Figure 1 depicts the proposed research model. The charity donating construct is wrapped in a dotted frame labelled as altruism to denote the close relation of these two concepts (i.e., altruism and charitable giving) in this study.

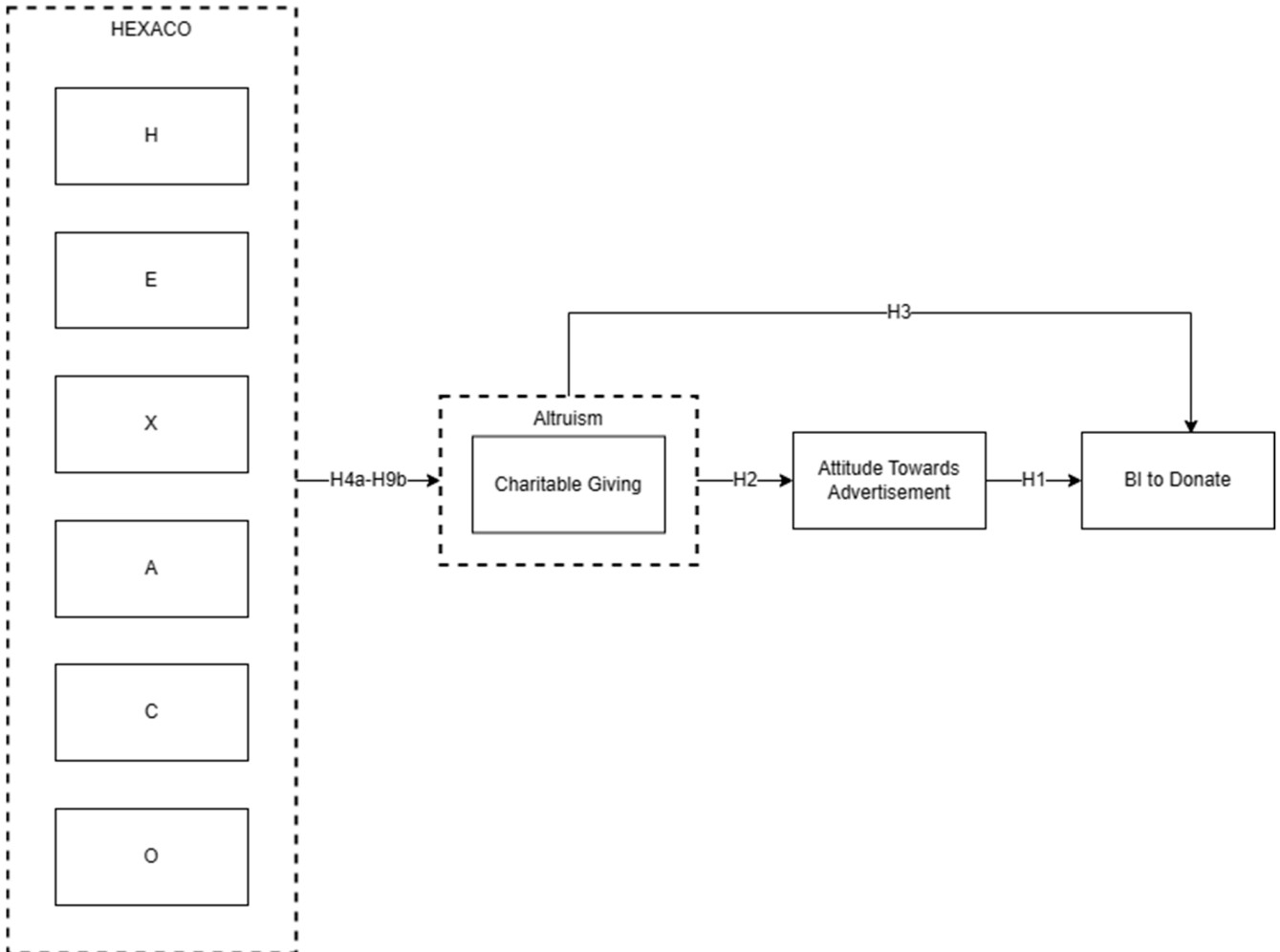

**Figure 1.** Research model. H1, H2, H3, H4a–H9b are the hypotheses described in Table 1.

Our study adds to the existing literature by investigating how people's altruistic dispositions impact charity donations, not only from the consumers' attitude towards advertising campaigns, but also by combining it with the personality traits that predominate among these dispositions. The approach is based on [57], whereas the present research introduces the dimension of personality. As an outcome, we examine each personality trait independently, beginning with the attitude towards advertising, and ending with altruism through charity giving, as well as the interconnected relationships arising from the aforementioned. Table 1 lists the hypotheses tested.

**Table 1.** Hypotheses.

| | Hypotheses |
|---|---|
| H1 | Attitude towards advertisement (AAD) has a direct positive effect on behavioral intention to donate (BI). |
| H2 | Attitude towards advertisement (AAD) has a direct positive effect on charitable giving (ACG). |
| H3 | Charitable giving (ACG) has a direct positive effect on behavioral intention to donate (BI). |
| H4a | Honesty-humility has a positive relationship with attitude towards advertisement (AAD). |
| H4b | Honesty-humility has a positive relationship with charitable giving (ACG). |
| H5a | Emotionality has a positive relationship with attitude towards advertisement (AAD). |
| H5b | Emotionality has a positive relationship with charitable giving (ACG). |
| H6a | Extraversion has a positive relationship with attitude towards advertisement (AAD). |
| H6b | Extraversion has a positive relationship with charitable giving (ACG). |
| H7a | Agreeableness has a positive relationship with attitude towards advertisement (AAD). |
| H7b | Agreeableness has a positive relationship with charitable giving (ACG). |
| H8a | Conscientiousness has a positive relationship with attitude towards advertisement (AAD). |
| H8b | Conscientiousness has a positive relationship with charitable giving (ACG). |
| H9a | Openness to experience has a positive relationship with attitude towards advertisement (AAD). |
| H9b | Openness to experience has a positive relationship with charitable giving (ACG). |

### 2.2. Methodology

After reviewing the literature, we developed a research model that comprises elements for the assessment of various behavioral and cognitive aspects that impact human behavior and shape fundamental beliefs for future donation. During the experimental procedure, a questionnaire comprising two parts was distributed to the participants. The first part of the questionnaire collected demographic data (gender, age, education, career, and other factors) and the 37 items of HEXACO-24. The second part exposed participants to a fictitious charity advertisement, followed by the 13 questions corresponding to the 3 factors of our research model, namely attitude towards advertisement, altruism (charitable giving) and behavioral intention to donate, as depicted in Appendix A.

Attitude towards the advertisement was measured with a four-item semantic differential scale, obtained and adapted from Holbrook and Batra [61] and Ranganathan and Henley [57]. The construct included the items of likeness (dislike/like the advertisement), favorable reaction (unfavorable/unfavorable), feelings towards the advertisement (negative/positive) and overall attitude (bad/good). To assess altruism, we used a 5-item measurement scale for charitable giving implemented by Johnson et al. [62] and adapted by Kaya et al. [58]. Behavioral intention to donate was measured with a four-item scale used by Coyle and Thorson [63] and adapted from Ranganathan and Henley [57]. Finally, personality traits were measured in six dimensions using the short version of the well-established HEXACO Personality Inventory-Revised [27], the brief HEXACO Inventory (BMI), which includes 24 items to measure the traits of honesty-humility, emotionality, extraversion, conscientiousness, agreeableness and openness to experience [64]. All items were measured in a 5-point Likert scale and, as the instrument was distributed to a Greek audience, were translated accordingly to ensure that the meaning of the questions remained intact for statistical validity in our analysis.

Our research was carried out utilizing online questionnaire distribution techniques that targeted specific recipients via email (primarily university students), and the general public accessible via social media platforms (Facebook, Instagram), to enable the involvement of individuals from various backgrounds and to achieve additional diversity within our sample. One such strategy we implemented was snowball sampling [65]. After participants completed the questionnaire, we encouraged them to recommend and distribute the instrument to others in their close environment (friends, relatives, etc.) to enhance participation. We additionally offered respondents the chance to enter a competition for a 100 EUR gift card as an incentive to participate in the survey. We collected data from 332 respondents. The descriptive statistics of the sample revealed that the sample was gender-balanced, with 47.9% of respondents being female and 52.1% male. In terms of age distribution, the age group "26–30" has the largest proportion with 36.1%, followed by

the age group "18–25" with 33.1%. The respondents' educational backgrounds are made up of graduates (33.2%) and undergraduate students (28.0%). Table 2 summarizes the demographic information of the sample.

**Table 2.** Sample profile.

|  |  | Frequency | Percentage |
|---|---|---|---|
| Gender | Male | 173 | 52.1% |
|  | Female | 159 | 47.9% |
| Age | 18–25 | 110 | 33.1% |
|  | 26–30 | 120 | 36.1% |
|  | 31–40 | 48 | 14.5% |
|  | 41–50 | 26 | 7.8% |
|  | 51–60 | 19 | 5.7% |
|  | 60+ | 9 | 2.7% |
| Education | High school graduate | 53 | 16.0% |
|  | Undergraduate student | 93 | 28.0% |
|  | Graduate | 107 | 32.2% |
|  | Postgraduate student | 43 | 13.0% |
|  | Postgraduate | 29 | 8.7% |
|  | PhD candidate | 1 | 0.3% |
|  | Doctoral | 1 | 0.3% |
|  | Other | 5 | 1.5% |

To overcome selection bias and identify causal effects, we implemented the "*one-group-post-test only research design*" specified by Shadish et al. [66] and adapted from [57]. This research design has no need for a control group. Since it allowed us to evaluate cause-and-effect correlations between factors, minimizing the impact of "*retrospective bias*" [67], and "*controlling for external variables*", we preferred an experimental approach [66]. To achieve our goal of studying the factors that influence charitable giving, and ensuring the reliability and validity of our data and experiment, we created a fictitious advertisement that depicted a fictitious charity and urged people to contribute financially to the charity organization's stated purpose ("Donate to give access to clean water to African populations"). The fictitious organization was chosen to eliminate user bias, as many individuals have opted to donate exclusively to organizations with specific goals that represent their ethical and philanthropic attitudes. The requested donation is 3 EUR per month or 36 EUR per year, which is considered to be a sufficiently small amount so as not to discourage a potential donor from contributing, since research has emphasized the relevance of the requested amount and its effect on charitable giving [44,68–71].

### 3. Results

Collected data were analyzed in SPSS using three techniques for statistical validation: sample descriptives, structural equation modeling (SEM), and regression analysis. For the validation data, we used composite reliability (CR), average variance extracted (AVE) tests, and Cronbach's alpha test. More specifically, this study utilizes SEM with maximum likelihood estimates as an analysis method, and data analysis was performed using SPSS Amos 26 [72]. The first step was to test the content, convergent, and discriminant validity of constructs using the measurement model, while the second step was to test the hypotheses with the structural model and the goodness of fit indices.

Furthermore, to explore the relationship between the HEXACO personality traits with AAD and AGC, a regression analysis was utilized. Results determined that honesty-humility (b = 0.392, SE = 0.191, t = 12,868, $p < 0.01$), agreeableness (b = 0.195, SE = 0.171, t = 14,779, $p < 0.01$), conscientiousness (b = 0.317, SE = 0.202, t = 11,726, $p < 0.01$), and openness to experience (b = 0.202, SE = 0.195, t = 14.07, $p < 0.01$), resulted in a significantly positive attitude towards AAD, while extraversion (b = −0.389, SE = 0.087, t = 48,221,

$p < 0.01$), and openness to experience (b = −0.458, SE = 0.094, t = 45,701, $p < 0.01$) resulted in a negative attitude towards ACG. Table 3 provides in detail the results of regression analysis.

**Table 3.** Regression analysis.

| Personality Trait | Hypotheses | R2 | F (Sig) | Beta | T (Sig) | SE | M | SD |
|---|---|---|---|---|---|---|---|---|
| Honesty-Humility | | | | | | | 3.9081 | 0.68309 |
| | H4a: Honesty-Humility-AAD | 0.154 | 59,877 (<0.01) | 0.392 | 12,868 (<0.01) | 0.191 | | |
| | H4b: Honesty-Humility-ACG | 0.131 | 49,961 (<0.01) | −0.363 | 44,782 (<0.01) | 0.102 | | |
| Emotionality | | | | | | | 2.6935 | 0.56524 |
| | H5a: Emotionality-AAD | 0.082 | 29,496 (<0.01) | −0.286 | 21,719 (<0.01) | 0.164 | | |
| | H5b: Emotionality-ACG | 0.1 | 36,807 (<0.01) | −0.317 | 25,509 (<0.01) | 0.086 | | |
| Extraversion | | | | | | | 3.5858 | 0.58957 |
| | H6a: Extraversion-AAD | 0.137 | 52,146 (<0.01) | 0.370 | 14,449 (<0.01) | 0.166 | | |
| | H6b: Extraversion-ACG | 0.151 | 58,808 (<0.01) | −0.389 | 48,221 (<0.01) | 0.087 | | |
| Agreeableness | | | | | | | 3.1318 | 0.57365 |
| | H7a: Agreeableness-AAD | 0.038 | 13,033 (<0.01) | 0.195 | 14,779 (<0.01) | 0.171 | | |
| | H7b: Agreeableness-ACG | 0.151 | 58,595 (0.01) | −0.388 | 43,999 (<0.01) | 0.085 | | |
| Conscientiousness | | | | | | | 3.5685 | 0.69987 |
| | H8a: Conscientiousness-AAD | 0.101 | 36,965 (<0.01) | 0.317 | 11,726 (<0.01) | 0.202 | | |
| | H8b: Conscientiousness-ACG | 0.215 | 90,380 (<0.01) | −0.464 | 44,687 (<0.01) | 0.1 | | |
| Openness to Experience | | | | | | | 3.4661 | 0.65655 |
| | H9a: Openness to Experience-AAD | 0.041 | 13,982 (<0.01) | 0.202 | 14.07 (<0.01) | 0.195 | | |
| | H9b: Openness to Experience-ACG | 0.210 | 87,703 (<0.01) | −0.458 | 45,701 (<0.01) | 0.094 | | |

### 3.1. Measurement Model

First, we assessed the reliability and validity of the measurement instrument using content, reliability, and convergent validity criteria. The content validity of our survey instrument was established in two ways. First, the constructs, along with their measures which are used in this study, were already validated in previous studies, as they were all adopted from the existing literature. Second, the results of the pre-test we undertook with subject-matter experts assured content validity of the survey instrument. For reliability of the scale, Cronbach's alpha, which is a common method used to measure the reliability and internal consistency of scales, was used [73]. Ursachi, et al. [74] suggested that the reliability of the scale is generally accepted if the value of Cronbach's alpha for each construct is equal or greater than 0.70. The constructs included within the study's model exhibit a high degree of internal consistency as the values of Cronbach's alpha ranged from 0.822 (ACG) to 0.9 (BI), as shown in Table 4. Composite reliability (CR) and average variance extracted (AVE) tests were conducted to measure convergent validity. Kline [75] suggested that the value of CR for each construct must exceed 0.70 while the value of the AVE must exceed 0.50 for the convergent validity to be assured. Our AVE are less than 0.5, but our CR is at the more-than-acceptable level of 0.6.

The CR and AVE values for the constructs included in the study model are all above acceptable levels. As such, content validity, reliability, and convergent validity of the measurement instrument are all satisfactorily met in this research.

### 3.2. Structural Model

As a first step, the exploratory factor analysis (EFA) was used to identify the factors underlying the variables of the questionnaire [76]. The varimax orthogonal factor rotation method was applied to minimize the number of variables that have high loadings on each factor. The results of the EFA suggested that one factor explained over 68% of the variance. The internal consistency is confirmed by calculating Cronbach's alpha and component reliability to test the instrument accuracy. The results of SEM analysis show the structural model, the estimates, and evaluation of the formulated hypotheses (Table 5), as well as the goodness of model fit indices [77]. The SEM fit index of RMSEA is marginally acceptable; but, as many studies suggest, this is a reasonable model combined with the overview of other indices [78,79] (Table 6). The results indicate that both BI and ACG are a negative

direct function of ADD (b = 0.06, *p* < 0.001, b = 0.12, *p* < 0.001, respectively). Finally, it was found that ACG has a strong direct effect on BI (b = 0.87, *p* < 0.001). Therefore, the indices of the goodness of fit are all acceptable.

**Table 4.** Results of reliability and convergent validity tests.

| Constructs | Items | Factor Loadings | AVE | CR | Cronbach |
|:---:|:---:|:---:|:---:|:---:|:---:|
| AAD | | | 0.44 | 0.89 | 0.891 |
| | AAD 1 | 0.770 | | | |
| | AAD 2 | 0.823 | | | |
| | AAD3 | 0.842 | | | |
| | AAD4 | 0.841 | | | |
| BI | | | 0.69 | 0.90 | 0.9 |
| | BI 1 | 0.778 | | | |
| | BI 2 | 0.906 | | | |
| | BI 3 | 0.875 | | | |
| | BI 4 | 0.772 | | | |
| ACG | | | 0.53 | 0.81 | 0.822 |
| | ACG2 | 0.541 | | | |
| | ACG3 | 0.849 | | | |
| | ACG5 | 0.614 | | | |
| | ACG1 | 0.876 | | | |

**Table 5.** Hypotheses testing results.

| Hypotheses | | | Estimates | Result (*p*-Value) |
|:---:|:---:|:---:|:---:|:---:|
| H1: AAD | <--> | BI | −0.06 | Not Supported (*p* < 0.001) |
| H2: AAD | <--> | ACG | −0.12 | Not Supported (*p* < 0.001) |
| H3: BI | <--> | ACG | 0.87 | Supported (*p* < 0.001) |

**Table 6.** Goodness of model fit indices.

| Goodness-of-Fit Indices | Value | Acceptable Values |
|:---:|:---:|:---:|
| TLI | 0.944 | >0.90 |
| RMSEA | 0.09 | <0.08 |
| GFI | 0.921 | >0.90 |
| CFI | 0.944 | >0.90 |
| NFI | 0.927 | >0.90 |

## 4. Discussion

The world's volatile current economy has not only affected the degree of charitable practices, but has also resulted in a degradation of the nature of human beings, making it more vital than ever to identify the associated factors driving compassion, empathy, and solidarity for fellow individuals. This research provides insight into how the factors of personality traits, altruistic behaviors, and humanitarian actions can positively influence citizens' behavioral intentions, allowing for a deeper understanding of the motives behind charitable giving.

We invoked factors related to the perceived value of the marketed message and organization, as well as psychometric variables such as people's altruistic nature and personality traits, in an endeavor to deeper understand the attitudes and behaviors that lead to charitable donations. We focused on the relationships established by the displayed advertisement, the altruistic tendency, and the final behavioral intention to donate while testing hypotheses H1–H3. We specifically investigated if there is a statistically significant positive relationship between the attitude toward advertisement (AAD), altruism via charitable giving (ACG), and the behavioral intention to contribute (BI). The hypothesis tests H1 and H2 are particularly interesting in our sample, since attitude towards advertisement (AAD), charitable giving (ACG) and behavioral intention (BI) were not supported, as evidence showed a

significant negative relationship between AAD with BI and ACG. Charitable organizations rely significantly on marketing campaigns that showcase their aims, with messages that inspire a call to action to achieve their goals and enhance the awareness and interest of consumers. The development of advertising campaigns is a crucial component not only to achieve maximum efficiency, but also to ensure that the message we are pushing is the suitable one based on the requirements of the charity and the values it represents [80–82]. As previously mentioned, emotional appeals are a vital aspect of the process, as well as the advertised content, as the results can differentiate based whether the appeal targets people's altruistic or egotistic nature. In our experiment, the displayed advertising was designed in such a manner that the emotional appeal was not overwhelming. Although it is of particular research interest to investigate the effect of emotions such as guilt, compassion, delight, and so on, the charitable donation amount was the dominant factor in our experiment [44,69,70]. Furthermore, despite the fact that we purposefully avoided user bias by designing a fictitious charity, research has shown that factors such as people's preferences, as well as other socio-political and external influences, can alter the perceived value of a charity organization. Nonetheless, our findings add to current research by underlining the relevance of individuals' altruistic character, since our findings revealed a statistically significant relationship between charitable giving (ACG) and behavioral intention to donate (BI) in H3. Thus, regardless of the advertisement's promoted altruistic ideals and willingness to contribute, people's original altruistic values and desire to contribute serve as a guideline in their search for social change and engagement. Prosocial practices in conjunction with humanitarian causes have a significant impact on altruistic dispositions, and can serve as motivators for individuals to do good and assist those in need in our society.

We attempted to integrate another dimension in our research, that of personality traits, to differentiate ourselves from socio-political factors such as the role of altruism and the advertisements we depict. The study of personality traits enables researchers to investigate the psychological and psychometric factors that may be utilized to increase the effectiveness of communicating advertising messages and, in our case, provide positive feedback to assure charity donations. Hypotheses H4a–H9b are assessed to determine how personality traits influence attitudes toward advertising, and how they relate to the concept of altruism. Five out of the six personality traits had a statistically significant positive relationship with attitude towards advertising (AAD). Only emotionality, however, did not result in a positive relationship with AAD. In terms of altruism, what is particularly intriguing is that none of the personality traits exhibited a statistically significant positive relationship with the concept of charitable giving. Our research results complement already-existing evidence that people with higher scores on traits such as honesty-humility, agreeableness, and extraversion exhibit positive behaviors toward charitable organization advertising messages, as well as sympathetic tendencies and externalized efforts to help and contribute to social causes through volunteering, donations, and so on [35,37,38]. We highlighted the importance of emotional appeals, which may have influenced users' altruistic attitudes, as it was not reflective of their personal values and views [44,59,83]. We intended to extend our research by deploying the HEXACO-60, since it takes a more thorough approach to the six personality traits and also has stronger reliability, which may have had a significant impact given that HEXACO-24 has not been widely established and validated by the scientific community [64]. Furthermore, concepts such as altruism are challenging to approach and quantify. We also plan to incorporate alternative measures of life satisfaction and subjective well-being, as it is probable that the concepts we utilized in our research did not accurately capture the users' altruistic behaviors and content. Our results demonstrate the need to delve more into why it is more usual to notice attitudinal changes rather than behavioral when investigating personality traits and future donations.

## 5. Conclusions

Humanitarian efforts are being made all around the world by non-governmental organizations, government institutions, charitable organizations, and so forth. The public regards their participation as a noble purpose. We support individuals in need, who are less fortunate than us or in tough situations, by making their life better and happier, easing some of their pain, and by providing charitable gifts to these people.

The present study aimed to explore the factors that influence charitable donations more deeply. For this purpose, through an experimental design, we incorporated numerous dimensions of the human condition as well as key elements of cognitive and psychological aspects related to altruism. We emphasized how personality traits are directly related to people's altruistic nature and attitude toward advertisement, as well as the overall impact they have on behavioral intention to donate. Our findings revealed that personality traits are strongly linked to attitudes towards advertisements, and we highlighted the significance of the relationship between charitable giving and behavioral intention.

It is regarded as vital to investigate methods of increasing donations. The complexities of this subject necessitate a more in-depth investigation of the variables that impact people's altruistic attitudes, and the extent to which they influence their inclinations for social service. From the perspective of organizations, this knowledge is a significant advantage in the strategic development of advertising campaigns to attract citizens. It is critical to raise awareness and sympathy for the philanthropic cause by providing relevant stimuli, either through an emotional appeal or a call for aid for a specific social purpose of the firm, in order to generate revenue. Research in this field allows us to gain a better insight into consumer behavior, but also to develop methods for optimizing consumer attitudes towards NPOs and charity organizations. Charitable attitudes can be difficult to address on a behavioral and cognitive level, since a multitude of factors could influence human behavioral intention. Personality traits delve into an individual's particular set of characteristics that influence the dynamics of how they perceive their environment. Thus, they represent an entirely novel perspective on the reasoning process that leads from social awakening via an advertising message to behavioral change in people's prosocial and altruistic tendencies, and therefore to charitable actions. On a practical level, rigorous design combined with personality research enables the development of more personalized messaging, with the objective to attract and target a broader customer group. Studying the effect of personalized messaging after advertisement exposure can potentially establish some fundamental recommendations for non-profits to guarantee that their advertising aims are accomplished and, ultimately, charitable donations and altruistic activities are maximized. The sampling technique and sample quality are two challenges raised by such research. Previous studies have focused on the general population, with a specific emphasis on the influence of different religions. As a substantial portion of the survey questionnaires were distributed within the university environment, our sample comprises a large fraction of primarily younger age groups with the profile of students. In the future, we intend to research non-student populations, although little to no effort has been made to study Greek residents' prosocial actions, attitudes, and dispositions. A significant portion of our study has been on the influence of personality traits and their impact on all aspects of human behavior that influence and engage with humans' altruistic and benevolent nature. In our study, we utilized HEXACO, a recognized personality test questionnaire. We adopted one of the short versions of HEXACO, namely HEXACO-24, to save time and minimize mental tiredness among our participants. In the current study, we did not account for the influence of certain traits through facets, as they offer a unique viewpoint and depiction of the personality, but we plan to pursue this in the future.

Considering past studies have focused on subjective well-being and life satisfaction, it would be intriguing to investigate how social conditions and socioeconomic appraisal impact Greek citizens and their lives. To our knowledge, no other research effort has been conducted employing a sample of the general Greek population, and it is one of the few research attempts to explore personality traits in this particular demographic. The

idea of quality of life is broad in scope, and is a multifaceted phenomenon. The process of measuring quality of life involves several mutually affecting aspects, which renders the task challenging. This realization occurs when the determination of quality of life is undertaken in a manner that allows for assessment and measurement of it. Nonetheless, we seek new methods and strategies for comprehending human nature and the elements that inspire individuals to be empathetic, compassionate, and charitable. Our findings not only provide a novel viewpoint on a person's cognitive and psychological condition, but also have practical implications for charity organizations, who are pushing harder than ever to improve donation behavior, raise awareness, and achieve social change.

**Author Contributions:** Conceptualization S.B. and M.R.; methodology, M.R. and S.B.; validation A.P.; writing—original draft preparation, S.B. and A.P.; writing—review and editing, M.R. and S.B.; supervision, M.R. All authors have read and agreed to the published version of the manuscript.

**Funding:** This research was financially supported by the Andreas Mentzelopoulos Foundation.

**Institutional Review Board Statement:** The study was conducted in accordance with the Declaration of Helsinki and approved by the Research Ethics Committee (REC) of the University of Patras (application no. 14045, date of approval 26 August 2022) for studies involving humans. The Committee reviewed the research protocol and concluded that it does not contravene the applicable legislation and complies with the standard acceptable rules of ethics in research and of research integrity as to the content and mode of conduct of the research.

**Informed Consent Statement:** Informed consent was obtained from all subjects involved in the study.

**Data Availability Statement:** Data analyzed in this article are available upon request by the corresponding author.

**Acknowledgments:** The authors would like to thank the anonymous respondents of the survey for their valuable contribution.

**Conflicts of Interest:** The authors declare no conflict of interest.

## Appendix A. Measurement Items Used for Data Collection

| Attitude towards advertisement (5-point semantic differential scale) | | |
|---|---|---|
| AAD1 | Like the ad (I dislike the ad—I like the ad) | |
| AAD2 | Favorable (I react unfavorably to the ad—I react favorably to the ad) | Originally Holbrook and Batra [61] |
| AAD3 | Positive (I feel negative toward the ad—I feel positive toward the ad) | Adapted from Ranganathan and Henley [57] |
| AAD4 | Advertisement is good (The ad is bad—The ad is good) | |
| Altruism (charitable giving) (5-point scale) | | |
| ACG1 | I have given money to a charity | |
| ACG2 | I have given a money to a stranger who needed it (or asked me for it) | |
| ACG3 | I have donated goods or clothes to a charity | Johnson et al. [62], and |
| ACG4 | I have pointed out a clerk's error (in a bank, at the market) in undercharging me for an item | adapted from Kaya et al. [58], |
| ACG5 | I have paid a little more to buy an item from a merchant who I felt deserved my support | |
| Behavioral intention to donate (5-point scale) | | |
| BI1 | It is very likely that I will donate money | |
| BI2 | I will donate money next time | Originally from Coyle and Thorson [63] |
| BI3 | I will definitely donate money | adapted from Ranganathan and Henley [57] |
| BI4 | I will recommend others to donate money | |

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
