# Peer review of "Impact of Personality Traits on Small Charitable Donations: The Role of Altruism and Attitude towards an Advertisement"

_societies, doi:10.3390/soc13060144_

Round 1

Reviewer 1 Report

I am sending some suggestions for an article.

Reviewer 2 Report

Overall, I have some ethical concerns about the deception of the respondents and where the charitable donations went. I also found some inconsistencies with regard to who the respondents were and how they were recruited. These concerns are highlighted in the attached document. Very well presented and written article, but some points of clarification are required as per the attached Word document.

Reviewer 3 Report

The article talks about charitable donations from the perspective of personality traits, without saying anything about the economic benefits of charitable activities. Is necessary to have a complete understanding of the effects of tax policies that affect monetary giving and volunteer labor. Milton Friedman is talking about social and economic objectives that are separate and distinct (Milton Friedman – Capitalism and Freedom). Another interesting work is signed by Porter, Michael E., and Mark R. Kramer. "Advantage." Creating and Sustaining Superior Performance.  Here a distinction is made between pure philanthropy and pure business in terms of social benefit and economic benefit. In fact, my main problem with this study is that it doesn't seem fair to talk about the influence of personality traits on charitable actions without knowing, even approximately, what weight personality has and what weight interest has in charitable acts. Not to mention at all a factor that could be decisive, I think, is a serious mistake.

Round 2

Reviewer 1 Report

Most of the changes introduced by the Author/Authors contributed to better readability and consistency of the article.

Author Response

Thank you for your review and feedback.

Reviewer 3 Report

In this situation it is very important to state more clearly that the work concerns small deeds of charity. Even the title should be slightly modified: "The Impact of Personality Traits on Small Charitable Donations: The Roles of Altruism and Attitude towards Advertisement". Also I belive the chapter 1.2. Charitable attitudes & advertisement does not match the authors' intentions. I think the study is about the influence of personality on acts of giving and not on consumer behaviour. Also if the article is looking at small acts of giving, in my opinion they are not related to advertising, organizations or government. I think the authors have engaged in too many directions of knowledge, some of which are unrelated. But at this stage the opinion of the scientific editor is most important.

Author Response

Thank you for your comments and suggestions. We have added the word Small (Charitable Donations) to the title according to your suggestion, we too believe that this makes the title more precise. Concerning section 1.2, the behavioral intention of users is our main topic and since we used an advertisement to this end, we consider it relevant to discuss related research on charity advertising. Overall the study implicates the topics of personality traits, charity advertising, and behavioral intention to donate thus we covered all of them in the related literature section.